METHODS

# AEGIS: Individual-based modeling of life history evolution

**Martin Bagic[1], Arian Šajina[2], William John Bradshaw[2], Dario Riccardo Valenzano**[1,3,4]*

**1** Leibniz Institute on Aging, Fritz Lipmann Institute (FLI), Jena, Germany, **2** Max Planck Institute for Biology of Ageing, Cologne, Germany,  **3** Balance of the Microverse Excellence Cluster, Jena, Germany, **4** Friedrich Schiller University, Jena, Germany

\* dario.valenzano@leibniz-fli.de

## Abstract

Nature presents a staggering diversity of life history strategies, ranging from rapid to slow onset of sexual maturity, short or long life, low or high number of offspring, and much more. Each species-specific life history trait reflects on the one hand specific adaptations to unique environments, e.g., nutrient availability, predation, parasite load, seasonality; and on the other hand, depends on past demographic constraints, such as population bottlenecks, migrations, etc. Studying life history diversity in nature and in the laboratory ultimately aims to identify the ecological, demographic, and intrinsic causes contributing to species-specific growth rate distributions, lifetime reproductive outcomes, as well as lifespans. However, for most species, we cannot rewind the evolutionary and demographic past to identify the causal chain of events leading to the present life history traits. We can infer past events only by sampling extant populations. *In silico* evolution has the advantage of providing complete time resolution for the events driving life history evolution and enables to directly test the impact of ecological and demographic variables on the evolution of life history traits. We developed AEGIS (*Aging of Evolving Genomes In Silico*), a software for individual-based modeling of life history trait evolution at the genotype and pheno-type level. AEGIS models life history traits evolution in response to a set of factors, including resource availability, extrinsic mortality induced by predators or parasites, different levels of germline mutation rates, population size, sexual vs. asexual repro-duction, and more. AEGIS serves as a powerful tool to model life history evolution and allows for parameter inference against ground truths. AEGIS can help gener-ate estimates for the evolution of different life history traits, such as age-dependent mortality and reproduction, in response to different selective pressures and intrinsic genetic constraints.

**Data availability statement:** All data underlying the findings reported in this manuscript were generated through simulations using the AEGIS software. The simulation configuration files, parameter settings, and plotting scripts necessary to reproduce the results are available in our GitHub repository: https://github.com/valenzano-lab/aegis. All relevant data and code are publicly accessible and fully documented to facilitate reproducibility.

**Funding:** The author(s) received no specific funding for this work.

**Competing interests:** The authors have declared that no competing interests exist.

## Author summary

Life history traits - such as lifespan, age at reproduction, and fertility - vary widely across species and environments and are shaped by evolutionary processes acting over long timescales. Because these processes cannot be directly observed, computational models play a key role in understanding how ecological pressures, demographic constraints, and genetics interact to shape life histories. Here, we introduce AEGIS, an individual-based simulation framework designed to study the evolution of life history traits and aging. In AEGIS, populations are composed of individuals with heritable genomes that determine age-specific survival and reproduction. Population-level patterns, including aging, lifespan variation, and demographic dynamics, emerge from interactions between individuals and their environment rather than being imposed by predefined equations. AEGIS allows users to explore how different mortality sources, resource limitation, reproductive strategies, and genetic architectures influence evolutionary outcomes. The framework produces detailed demographic, phenotypic, and genetic outputs, enabling direct analysis of both population averages and individual variation. By combining flexibility, transparency, and reproducibility, AEGIS provides a general platform for investigating the evolutionary biology of aging and life history in silico.

## Introduction

Life history traits display broad natural variation both within and between species [1,2]. Species-specific life history traits evolve to optimize life-long reproduction [3]. The field of life history evolution investigates the evolutionary forces that explain why species reproduce repeatedly throughout adult life (iteroparity) or in individual episodes (semelparity), why species have short or long lifespans, why they grow fast or slow, etc. Different life history traits, such as growth, reproduction, and survival, are tightly interdependent as they rely upon the access to shared energy and metabolic resources [4].

Furthermore, life history traits can be genetically interdependent, e.g., via pleiotropy, i.e., by sharing the same genetic determinants [5]. Given their metabolic and genetic interdependence, often life history traits trade off, both metabolically, as well as evolutionarily [6,7]. Ecological constraints are important in shaping life history evolution, selecting traits based on their fitness outcome. Directional selection can favor life history traits that provide optimal solutions, e.g., in stable or predictable environments [8]. Ecological fluctuations and environmental heterogeneity, on the other hand, can favor mixed strategies, where no optimal life history trait solution exists even within a species [9]. Combinations of different life history traits – e.g., slow growth and long lifespan vs. fast growth and short lifespan – can therefore co-evolve even within the same species in response to varying environmental conditions [1,10].

While life history trait evolution has been largely explored at the macro-evolutionary scale, we still lack a deep understanding of life history trait evolution at the micro-evolutionary scale [9]. We still lack a general theory to explain the impact of germline and somatic mutation rate, recombination rate, sexual vs. asexual reproduction modality, population size in the evolution of different life history traits, such as age-dependent survival and reproduction.

While the exploration of the evolution of life history traits under different ecological constraints has largely relied on experimental work in wild and laboratory species, modeling provides a formidable tool to isolate the causal factors affecting – alone or in combination – life history trait evolution.

Despite decades of research on the evolutionary causes of aging, there are still several important open questions. It is unclear under which environmental conditions negligible senescence can evolve, or when lifespan compression can be favored. The genetic basis of aging, including lifespan heritability and the relative influence of intrinsic versus extrinsic factors, is still poorly understood. Tradeoffs between reproduction and survival, along with ecological pressures such as infection, predation, and environmental variability, are central to shaping senescence, yet their interplay remains difficult to disentangle. Whether aging is adaptive, and how reproductive strategies (e.g., sexual vs. asexual, oviparity vs. viviparity) modulate lifespan evolution, are also open questions. These challenges highlight the need for integrative theoretical frameworks that bridge ecology, genetics, and evolutionary biology – models capable not only of reconciling disparate findings, but also of accommodating biological complexity, relaxing simplifying assumptions, and capturing inter-individual variability.

Optimality models and population matrix models constitute the core of life history theory, owing to their explanatory and predictive power. However, they present several technical limitations, such as difficulty in handling stochasticity, environmental variability, complex genetics and within-population diversity.

Individual-based models (IBMs) [11,12] can overcome these limitations. Furthermore, IBMs provide a conceptual advantage as bottom-up models – observed outcomes are emergent instead of being assumed and built into the model. Additionally, in comparison to analytical methods, IBMs are easier to use and thus might be more broadly accessible.

IBMs have been widely employed across disciplines [13]. In the field of life history evolution, IBMs are often built *ad hoc* [14–27], sometimes based on conceptual models such as the Penna model [28,29] or, rarely, using generalist agent-based model (ABM) toolkits [13,30–32] such as MASON [33] and IBMPopSim [34]. More general modeling frameworks exist, such as NetLogo [35] and Mesa [36] but we could not find examples of their use in the field of evolutionary biology of aging. In the fields of ecology and evolution more broadly, in addition to a number of custom-made individual-based models, there are highly utilized domain-specific IBM packages, such as DEB-IBM [37] (a model of dynamic energy budget theory) and SLiM [38] (a population genetic simulation framework). However, currently, there is no domain-specific software for running IBMs to model evolution of aging.

Here we present AEGIS – a cross-platform IBM modeling software for studying evolution of life history and aging. It is made to be highly performant, reproducible, extensible, and accessible with a GUI featuring documentation, customization and analysis. The current feature set makes it especially useful for studying how life history evolves under diverse environments, mortality structures, intrinsic constraints, tradeoffs and genetic architectures, overlapping and non-overlapping generations, and population dynamics. It can also be used as an aid during hypothesis generation (to take on a different perspective, help think systemically, and establish theoretical expectations), experimental planning (to refine the working model and identify important factors), and post-experimental analysis (to test generality, improve reproducibility, and corroborate and contextualize findings in theory). Furthermore, AEGIS is suited for educational purposes, able to simulate, re-derive and study classical topics, e.g., predator-prey dynamics, epidemiological models, genetic drift – and explore custom scenarios, focusing on relevant concepts and processes rather than the technical aspects. Furthermore, AEGIS comes with built-in analysis and visualization, and a low barrier of entry via webserver in addition to an installation on a local machine.

## Methods

The model description follows the ODD (Overview, Design concepts, Details) protocol for describing individual- and agent-based models, as proposed by Grimm et al. [39].

### 1. Purpose and patterns

The general purpose of AEGIS is to simulate evolution of life history and aging under diverse environments, mortality structures, intrinsic constraints, tradeoffs/ genetic architectures, and population dynamics. The model's suitability can be tested by its ability to reproduce various life history and aging patterns, including the evolution of age-dependent hazard rates (e.g., manifested as increasing intrinsic mortality with age), mortality plateaus, evolution of reproductive aging (manifested as decreasing or varying fertility with age), oscillating population sizes, evolution of the mutation rate [40], and convergent evolution of life history (Results).

### 2. Entities, state variables, and scales

The model contains two entities – *environment* and *individual*. There is one instance of the environment entity and many instances of the individual entity, constituting the population.

The central subject of interest is the evolution of individuals inhabiting a shared environment (Table 1).

### 3. Process overview and scheduling

This section outlines model processes, the order of their execution and the rationale for their inclusion.

The model runs in three phases: initialization, simulation and termination.

During the initialization phase, the population is created (details in section 5).

During the simulation phase the population evolves. This phase runs in steps whose number is defined by an input parameter, by default set to 1 million. During each step, a list of processes is executed: some, conceptually concerning the environment (e.g., growth of the predator population), indirectly affect individuals; others concern the individuals directly (e.g., individuals mating). Ultimately, all processes together impact the survival and/or reproduction of individuals in the population. The order of processes given default parameters is illustrated in Fig 2.

Metadata about the simulation run is recorded during the termination phase.

**Table 1. State variables and scales of the two model entities (environment and individual), following the ODD protocol.**

| | |
|---|---|
| State variables of the *individual* entity | • Life phase. Egg or living individual.<br>• Age. An integer value, expressed in the time units of simulation steps.<br>• Genome. A bit string (a vector of 1's and 0's) encoding heritable information. Stable across life (no somatic mutations).<br>• Infection status. Healthy or infected.<br>• Intrinsic mortality rate. If age-dependent, a vector of real numbers, otherwise a constant.<br>• Fertility rate. If age-dependent, a vector of real numbers, otherwise a constant.<br>• Mutation rate. If age-dependent, a vector of real numbers, otherwise a constant. |
| State variables of the *environment* entity | • Step.<br>• Starvation history. Number of consecutive steps that the population size exceeded the carrying capacity of the environment.<br>• Number of predators. |
| Time | Time is simulated as discrete steps. |
| Space | Space is not simulated. |

## 1. Mortality

Individuals can die from different causes, modeled as one of the five processes listed below.

**1.1 Intrinsic mortality.** Intrinsic (genetic) mortality covers mortality that is truly independent of all external factors, including other individuals. Intrinsic mortality depends entirely on age and genetics and can evolve throughout the simulation. Mapping the evolution of age-dependent intrinsic mortality is one of the main features of AEGIS and can help dissect how ecology and demography can shape aging and lifespan at the individual and population level.

**1.2 Abiotic mortality.** Abiotic mortality models death by extrinsic factors that have a periodic character (e.g., temperature, water availability and other seasonal or weather-related factors). This source of mortality is independent of age and genetics. It is modeled using periodic wave functions parameterized by user input.

**1.3 Infection mortality.** Infection mortality models death by communicable disease. The parameterization enables users to customize various aspects of the model, including the transmissibility of the infection, its fatality rate, and the recovery rate. Infection mortality is independent of age and genetics, a property that cannot be modified by the user in the current version of AEGIS.

**1.4 Predation mortality.** Predation mortality models death by predators. Through parameterization, users can customize the intrinsic growth rate of the predator population and the vulnerability of simulated individuals to predation. Predation mortality is independent of age and genetics, a property that cannot be modified by the user in the current version of AEGIS.

**1.5 Starvation mortality.** Starvation mortality models death by starvation experienced by populations that exceed carrying capacity of the environment. This source of mortality is independent of genetics. Using default parameters, it is independent of age as well. However, it can be set to be dependent on age, for instance, so that older individuals are more likely to die due to starvation. Resource consumption is equal for all individuals and all ages. Since population growth is not "pre-set" upon initialization and depends on individual lifespan, reproduction, etc., starvation mortality is an emerging parameter that entirely depends on the contingencies of the simulation dynamics.

This source of mortality is useful for preventing computational overload caused by overpopulation. Mortality rate depends on the extent of population overshoot, with two starvation models implemented: *instantaneous* and *gradual* death. In the instantaneous model, fully fed individuals survive while those exceeding the carrying capacity die immediately. In the shared model, resources are evenly distributed, and all individuals face an equal mortality risk determined by a susceptibility parameter $m$. Mortality increases exponentially with each consecutive step of starvation $t$, with survival decaying as $(1 - m)^t$.

## 2. Reproduction

Reproduction is critical for keeping the population alive and evolving. The probability of reproducing is dependent on age, genetics, and availability of mates if reproduction is sexual. Each reproducing individual can produce at most one offspring during one reproductive event. Each individual can reproduce at every simulation step beyond the age corresponding to sexual maturity.

Non-customizable reproduction mechanics include random mating and no parental care.

Customizable reproduction mechanics include asexual vs. sexual reproduction (including assortment), recombination rate, germline mutation rate (parameterized or evolved), viviparity vs. oviparity (including incubation time), age at maturity, age at menopause.

Under viviparity, generations are overlapping. If oviparity is activated, there will be an *egg* life stage during which the developing individuals are invulnerable to all mortality sources. Under oviparity, generations can be overlapping or non-overlapping, depending on user input.

### 3. Aging

This process increments the *age* variable of each individual, resulting in a + 1 age at each discrete simulation step.

### 4. Environmental drift

As a default setting, environmental drift is deactivated, but it can be turned on. Conceptually, environmental drift simulates long-term environmental change such as climate change, resource depletion, pollution, etc.

The main use of environmental drift is to allow the population to keep evolving *adaptively*. When the environment does not change, the fitness landscape is static. After the initialization of each simulation, populations undergo an evolutionary phase, away from the initial conditions, in response to the initialization parameters, such as population size, reproduction modality, mutation rate, etc. Once populations approach a stable state, natural selection acts mostly to purify new detrimental mutations, since the environment is not programmed to change. However, if environmental drift is activated, the environment undergoes shifts, which might change accordingly the fitness landscape over time. With active environmental drift, populations keep evolving adaptively, following their fitness peak.

## Computational order

### Order of processes

Under the default parameters, mortality processes are scheduled as illustrated in Fig 2, in the following order: intrinsic mortality, abiotic mortality, infection, predation, starvation, reproduction, aging, and environmental drift. The order of mortality processes can be customized by the user.

Although the order in which mortality sources are applied will not affect the total number of deaths, it can impact death structure and thus affect demography.

For example, in a population of 1000 individuals, a first source with 40% mortality results in 400 deaths, and a second source with 80% mortality causes 480 deaths, as it acts on the remaining 600. The observed, population-level mortality rates are then 40% and 48% (rather than 40% and 80%). The order of these mortality events does not change the total number of deaths (880 in both cases), corresponding to a total mortality rate of 88%. However, it does alter the death structure (40% + 48% vs. 8% + 80%), which may affect the population demography if, for example, one source of mortality impacts all age groups equally, while the other disproportionately affects older individuals.

### Order of individuals

The order in which individuals are simulated can impact life history outcomes, similar to how process computation order influences the effects of each process. In many individual-based models, outcomes for individuals are computed sequentially—survival, reproduction, and other processes are simulated for one individual at a time, moving through the population until all individuals have been processed. The order of individuals in this sequential approach matters, as earlier computations can influence the outcomes of individuals simulated later. For example, the first individual seeking a mate has access to a larger pool of potential mates, while the last may face limited or no mating opportunities, potentially reducing fertility. If the order is not randomized but based on traits like age, this can introduce bias into the simulation.

In contrast, AEGIS uses parallel computation, where processes are computed simultaneously for all individuals. This removes the impact of one individual's life history outcomes on another's within the same simulation step.

### 4. Design concepts

**Emergence.** Emergent model outcomes are those not directly predictable nor imposed by a programmed rule but result from complex interactions of the model components. For example, in AEGIS, age at maturity is set by the user and thus not emergent, while lifespan is emergent as it is not set by the user nor fixed in the AEGIS code.

In AEGIS, the following outcomes can be emergent: evolvable phenotypes (survival, reproduction and mutation rates), population dynamics (population size, age structure, and dynamics of starvation, predation and infection) as well as distribution of genetic variants.

Some life history traits, such as age at maturity, are set by the user and are therefore not emergent in the current version of AEGIS. However, the current codebase is designed to support making these traits evolvable in the future.

**Interaction.** Individuals do not interact with each other directly.

However, indirect interaction occurs through the environment because each individual affects the predation, infection and starvation mortality of other individuals by modifying the presence of predators, infection and resources. In sexually reproducing populations, indirect interaction also occurs during mate pairing: although reproduction is implemented in parallel, each individual can mate at most once per simulation step. Once paired, individuals are removed from the mating pool for that time step, thereby reducing mate availability for others.

**Stochasticity.** Pseudorandom number generation is used in multiple processes.

- Composition of initialized genomes.

- Computing mortality outcome from various sources.

- Computing infection status (new infection or recovery).

- Computing reproductive success.

- Mating choice.

- Offspring genetics (three stochastic processes – recombination, mutation and assortment).

- Environmental drift.

- Population genetic estimates on population samples.

Random seeds can be applied to the model configuration to ensure reproducibility.

## 5. Experimental design: Parameterization, initialization and termination

Experiments in AEGIS follow a simple experimental design that specifies a baseline simulation and one or more comparative simulations, and that considers three components – parameterization, initialization and termination (Fig 1). After running comparative simulations, users can compare the evolved life history traits across these simulations and causally link their differences to variations in evolutionary conditions.

The first step is to appropriately parameterize simulations, i.e., adjust default parameter values, to emulate the evolutionary conditions of interest. Each evolutionary experiment requires at least two parameter sets – one representing the baseline conditions, and one or more capturing comparative evolutionary scenarios of interest. The built-in parameters and their explanations are listed in S1 File Supplementary Information.

The next step to consider is initialization. AEGIS does not require empirical life history traits as input; instead, it generates baseline populations *de novo*. While users can specify initial population-level life histories through adjustable parameters—useful when investigating evolutionary dynamics from a specific phenotypic starting point, such as assessing whether a non-aging population can maintain its non-aging phenotype—in most cases, a burn-in simulation is recommended. A burn-in simulation functions as the control condition against which experimental runs can be meaningfully compared.

Not using a burn-in simulation as a baseline for comparison may result in biased interpretations. For instance, if a non-aging survival phenotype is set as the initial condition, and multiple comparative scenarios are simulated where the user observes that the lifespan decreases at varying rates, it would be incorrect to attribute the observed reductions in

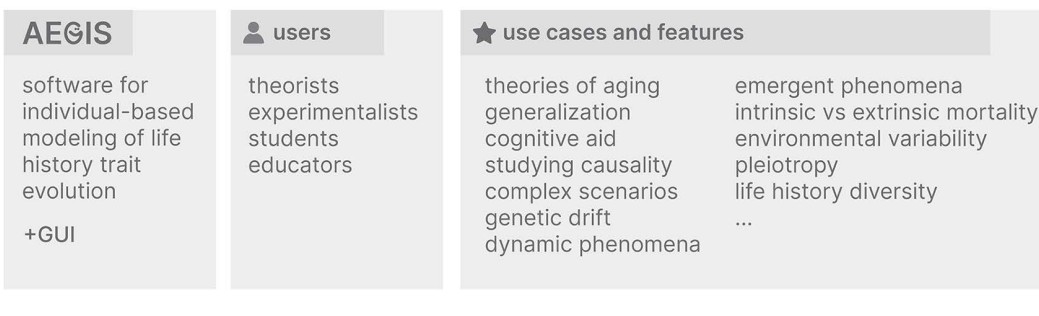

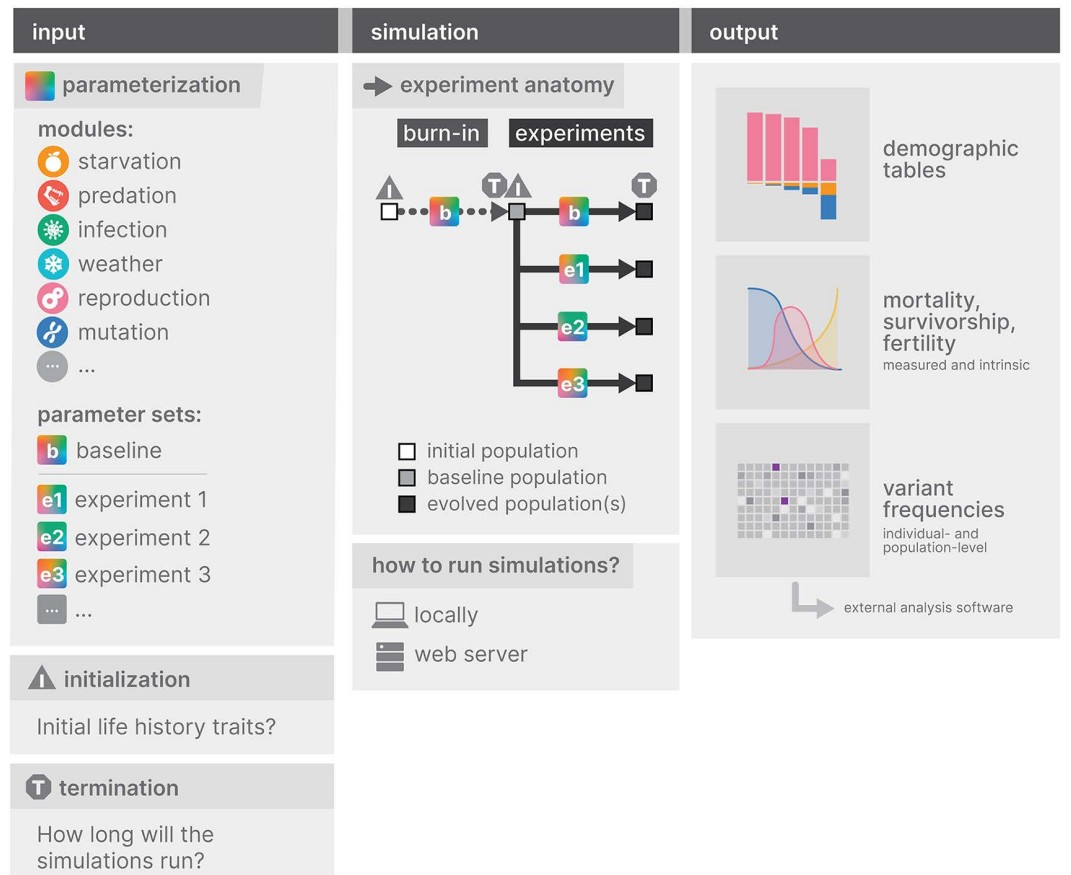

**Fig 1. Experimental design framework in AEGIS.** AEGIS simulations are defined by three core components—parameterization, initialization, and termination—applied to a baseline simulation and one or more comparative scenarios. Differences in evolved life-history traits can then be causally attributed to changes in evolutionary conditions.

lifespan solely to the varied parameter. This is because the interpretation of lifespan reduction is relative to and contingent on the high initial survival condition. In a population with a low initial survival, the opposite might be observed, i.e., population lifespan might increase. Thus, the same parameterization can lead to different interpretations depending on the choice of the comparative baseline. The burn-in simulation produces a population that has reached equilibrium based on the values of all parameters that will remain unchanged in subsequent experimental runs. This ensures that any evolved differences during experimental runs can be unambiguously attributed to interventions in parameter values of those same

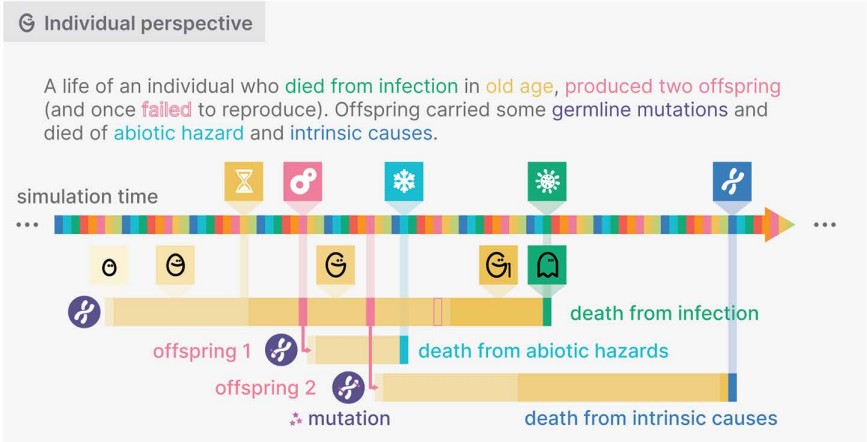

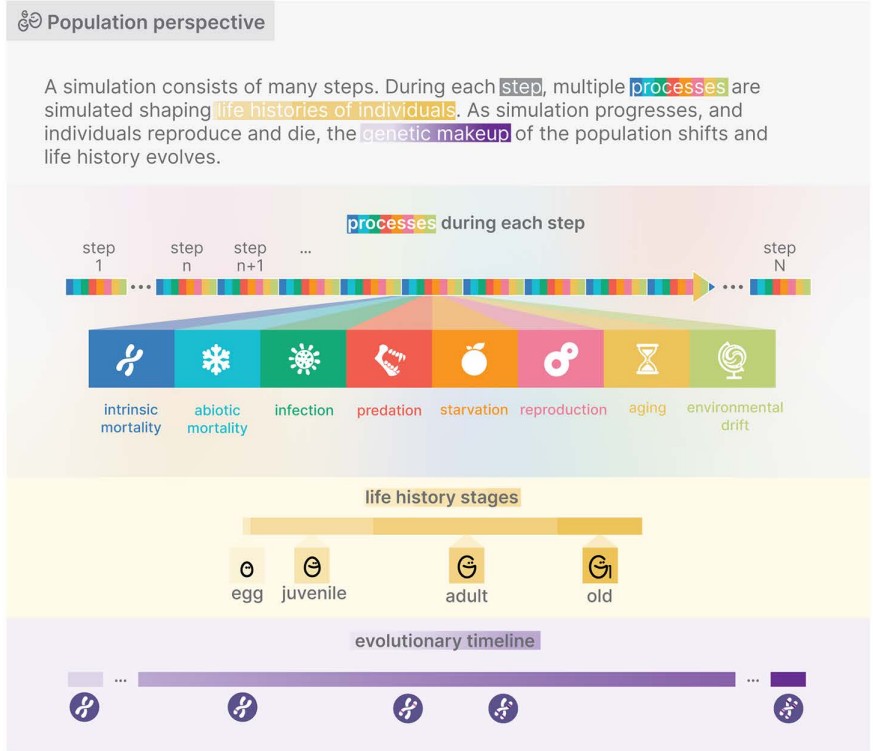

**Fig 2. Methodological summary of AEGIS.** During each simulation step, a predefined sequence of environmental and individual-level processes is executed, jointly determining survival and reproduction. The default order of these processes is shown.

experimental runs. Analogously, at the genomic level, burn-in simulations help establish a mutation-selection-drift balance, which is critical when evolutionary dynamics are studied.

Finally, users need to decide on the simulation termination point, i.e., the number of steps for the simulation to run. The total number of simulation steps represents a tradeoff: fewer steps result in faster simulations and reduced storage use but may provide insufficient time to observe relevant evolutionary outcomes or achieve equilibrium. To determine if equilibrium has been reached, it is best to compare the life history traits of interest over time – if no significant changes

are observed after a considerable period, the population has likely reached equilibrium (S1 File Supplementary Information).

### 6. Input data

Input data consists of the list of customized parameters and, optionally, a file containing a pre-evolved population. Input data can be entered via the graphical user interface, as a text file in YML format when AEGIS is used via a terminal, or as a dictionary when AEGIS is used via scripting.

### 7. Output data

Simulation outputs in AEGIS are organized into three main categories: demographic records, life-history traits, and variant frequencies.

Demographic tables include longitudinal records of population age structure, age-specific death distributions by cause, and birth distributions by parental age. From these, population-level life-history measures (e.g., median lifespan and life expectancy) can be derived, many of which are directly included in the output and visualized in the GUI. Demographic data are also amenable to further analysis using life history theory tools such as Leslie matrices.

In addition to population-level data, AEGIS records individual life-history trajectories, including birth time, age at death, and reproductive output. Importantly, it also tracks intrinsic survival and reproduction probabilities, which reflect biological potential independent of realized outcomes, allowing users to disentangle genetic from environmental contributions to life history.

Finally, AEGIS provides pseudogenomic data for each individual, enabling exploration of the genetic architecture underlying both individual- and population-level life histories. This resolution supports evolutionary analyses of trait dynamics shaped by selection and drift. With appropriate preprocessing, standard population genetic methods can be applied to investigate selective sweeps, linkage disequilibrium, and allele frequency change.

## Results

As a tool for studying life history evolution, we designed AEGIS to be able to simulate a population of individuals with age-specific survival and reproduction rates, which are mutable and heritable. AEGIS facilitates the study of life history evolution across both short (spanning several generations) and long time scales (beyond one million generations) under diverse environmental conditions and intrinsic constraints. AEGIS generates comprehensive demographic and phenotypic data at both the individual and population levels, encompassing both observed and intrinsic attributes, as well as genomic data critical for exploring the evolutionary mechanisms driving life history evolution. Given its modeling capabilities and ease of use, AEGIS empowers a research community from diverse technical backgrounds to tackle fundamental questions in evolution of aging.

We start by introducing the main features of the AEGIS framework (Figs 3–6). Subsequently, we showcase how to design and run an evolutionary experiment using AEGIS by recreating the Rose's experiment [41] – a classical experiment in experimental evolution of aging that aims to test whether delayed reproduction can over generations lead to evolution of delayed aging (Fig 7).

### AEGIS generates emergent demographics and life history traits

First, we ask whether aging can evolve spontaneously starting from a non-aging population within the AEGIS framework – in particular, we study how mortality and fertility curves are shaped during the simulation. Since our aim here is to demonstrate capabilities of AEGIS, rather than to replicate life history evolution in a particular scenario or species, we ran following simulations using default parameters (S1 File Supplementary information: Default parameter values). Here, default parameters broadly model humans – e.g., simulated populations reproduce sexually after a maturation phase, they

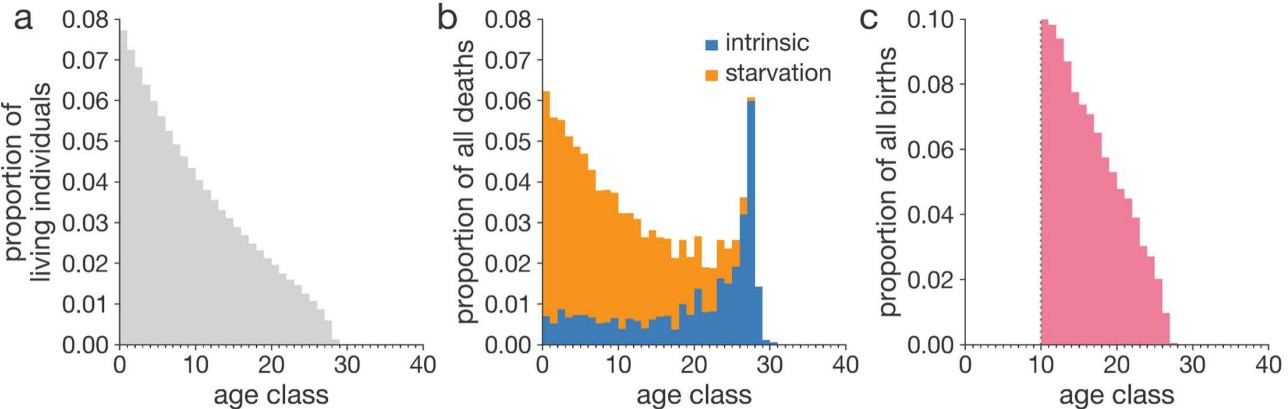

**Fig 3. Demographic traits of an evolved population.** In this example, we initialized the population with a non-aging (no increase in intrinsic mortality nor decrease in fertility with age), sexually reproducing population and ran the simulation for three million steps. a, Age structure of the evolved population. b, Resulting emergent distribution of deaths by cause and age class. Death by starvation is caused by the population exceeding the carrying capacity of the environment. Intrinsic death is caused by the inherited genetic makeup. c, Resulting emergent distribution of births by age class. Individuals are defined as mature from age class 10. Demographic tables reveal a sustainable demography with more young individuals than old. Consequently, the young individuals are shaping the demographic tables as they contribute more to the pool of the next generation but also to the total number of deaths in the population (Fig 3b). Demographic data has been collected for 1000 simulation steps, starting with the simulation step 2,900,000.

exhibit overlapping populations – with some simplifications such as random mating and no simulation of parental investment (S1 File Supplementary Information).

We computed age-specific mortality and fertility by dividing the number of births (Fig 3b) and deaths (Fig 3c) for each age class (Fig 3a) over the total number of births and deaths for a given simulation time interval. Compared to the initial non-aging phenotypes, characterized by age-flat mortality and fertility, evolved populations display age-increasing mortality (Fig 4b) and age-decreasing fertility (Fig 4c), indicating that somatic and reproductive aging can spontaneously evolve in AEGIS. We detect these age-specific patterns after 250 generations (S2d-e Fig; 5,000 simulation steps). Simulated populations reach an equilibrium by generation 20,000 (S2g-h Fig; 240,000 steps), after which they are maintained until generation 200,000 (Fig 3; 3 million steps) when we finally terminated them.

Notably, in the current experiment, mortality increases unevenly across different ages. Several species [42], including humans [43], have been documented to exhibit a late-life plateau in mortality rates, suggesting the intriguing possibility that aging may decelerate or even halt in later life. Within AEGIS, we observe a similar flattening of the evolved mortality curve in late life (Fig 4b), akin to the mortality plateau documented in extreme human survivors, indicating that AEGIS could be used to study factors shaping the evolution of mortality deceleration.

Mortality plateaus are often attributed to extrinsic factors, such as reduced exposure to hazards, leaving open the question of whether mortality deceleration also occurs on the intrinsic, physiological level. Since AEGIS models intrinsic causes of death separately from extrinsic causes, such as starvation, it can disentangle intrinsic mortality from extrinsic (Fig 4e) enabling us to study deceleration of mortality at the intrinsic level as well.

Much like studying late-life mortality trajectories, AEGIS serves as a tool for investigating at what age aging starts. Molecular evidence hints that aging might start as early as birth or during early life [44], while theoretical models of resource allocation suggest different ages as starting points [45], often highlighting sexual maturity as an inflection point in life history. In our example, evolved population starts "aging", measured as the acceleration of age-specific population mortality, at the onset of sexual maturity. To note, this result is not "imposed" as a parameter, but is rather emerging during simulations. More specifically, even though observed mortality rates seem to rise a few age classes after reaching maturity (Fig 4b), analyzing the intrinsic mortality data we found that aging starts right at the point of maturity itself (Fig 4e).

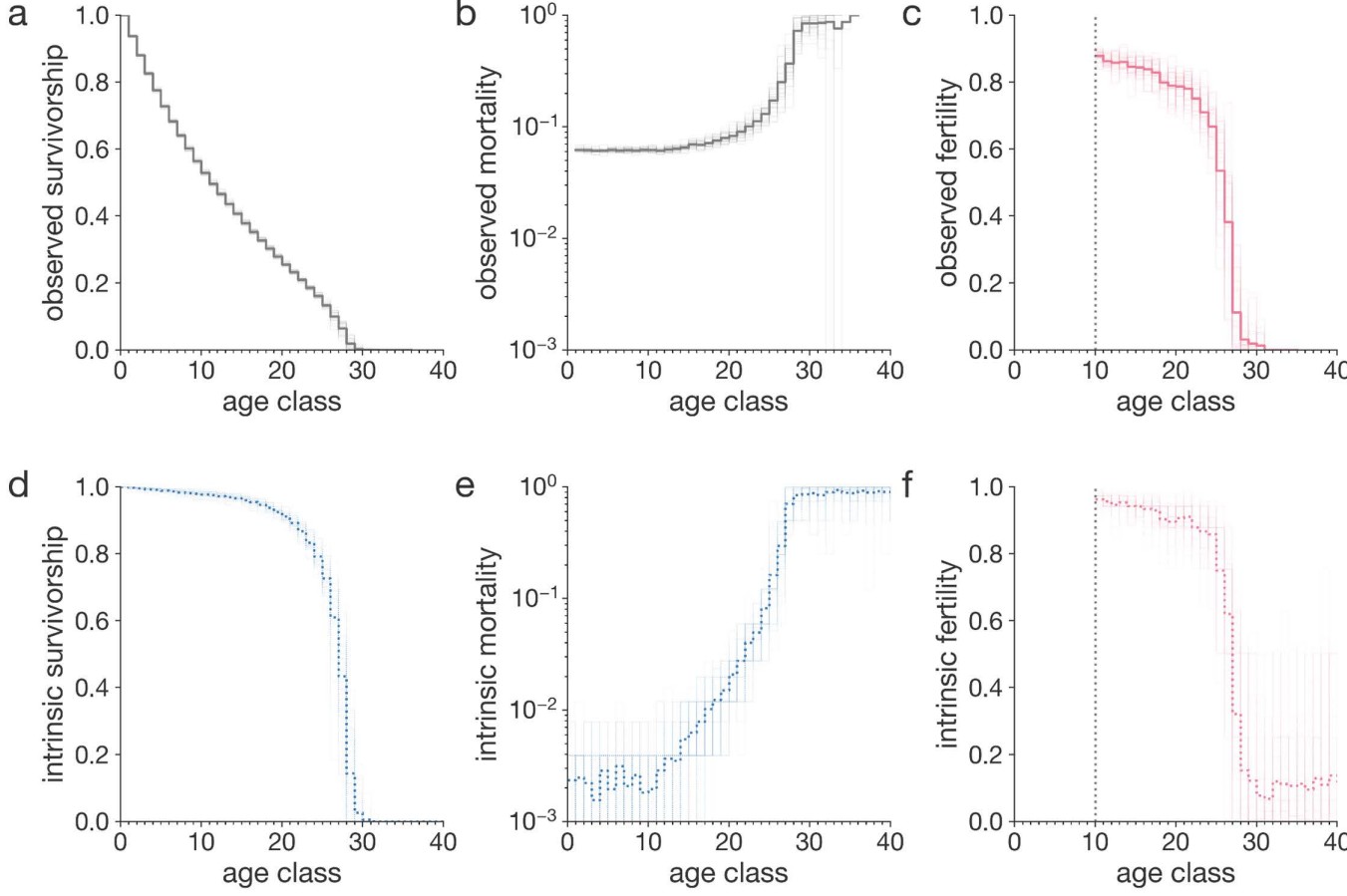

**Fig 4. Life history traits of evolved populations.** The life history curves shown are based on 100 simulations we ran for three million steps. Thick curves are simulation-averaged observations. Thin lines are simulation-specific observations. All curves are population averages, i.e., averages over all individuals in a population. a, Observed survivorship. b, Observed mortality. c, Observed fertility. d, Intrinsic survivorship, i.e., survivorship that would be observed if there was only intrinsic mortality and no extrinsic mortality. e, Intrinsic mortality. Mortality plateau is reached around age class 28. f, Intrinsic fertility. Observed life history traits were computed from the demographic tables. Intrinsic life history traits were recorded at step 3 million. Note that fertility is scaled by a factor of 2. The maximum attainable probability to reproduce for each individual is 50% per step.

A more sensitive detection of the onset of aging is possible looking at intrinsic mortality since the baseline is lower than the baseline of total mortality. We hypothesize that such an aging "schedule" is driven by strong selective pressures on variants that increase survival in the pre-reproductive stage. However, not all simulations in AEGIS will necessarily exhibit a flat pre-reproductive mortality. Increasing or decreasing pre-reproductive mortality (S1 File Supplementary Information: Juvenile mortality) might evolve under specific evolutionary scenarios.

## AEGIS reveals within-population life history variation

Even though life history traits are often analyzed and represented at the population level, substantial variation in survival and reproductive outcomes may exist between subpopulations or even individuals. One approach to examining these differences is through the analysis of survivorship curves, which can exhibit considerable variation across species [46]. More specifically, the so-called *rectangularization* of the survivorship curve [47] can indicate the level of life history diversity, driven by either genetic differences or extrinsic factors.

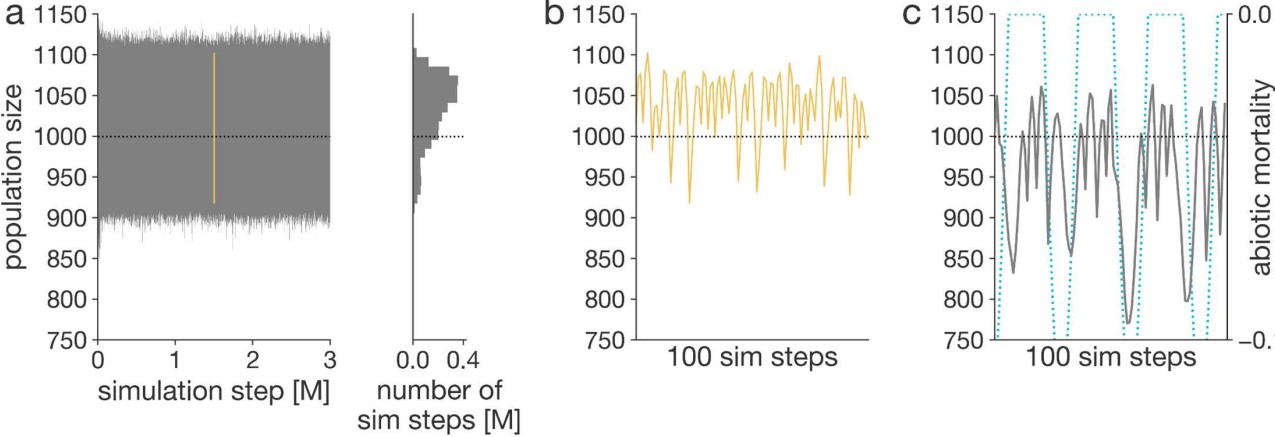

**Fig 5. Population size fluctuations during the simulation.** Population sizes during the simulations. Carrying capacity is set to 1000 (horizontal dashed line). Population size is recorded immediately after reproduction. a-b, Population size over the course of the simulation based on default parameters. The arrow points to the yellow subsection, zoomed in on in panel **b.** The accompanying histogram shows the distribution of observed population sizes. b, Population size over a hundred-step interval in the middle of the simulation. c, Population size during another simulation with imposed periodic weather-related mortality (blue dashed line), showing frequent oscillations and intermittent sharp drops.

In the simulations shown above, survivorship curves display a convex shape (Fig 4a, type II-III [48]), corresponding to high variance in age at death and rarity of survival into late life (Fig 2b). We hypothesized that this pattern was driven by high extrinsic mortality due to starvation (Fig 3b) generated by intense reproduction and subsequent overcrowding. Leveraging the ability of AEGIS to disentangle extrinsic factors of mortality, such as starvation, from genetically encoded intrinsic factors, we tested our hypothesis by replotting the survivorship curves considering only environment-independent intrinsic mortality. This differentiating analysis revealed a markedly rectangular survivorship curve (Fig 4d, type I [48]), implying that the intrinsic survival potential among individuals is highly homogenous and that extrinsic stochastic hazard drives lifespan variation. This finding may parallel the secular trend of rectangularization documented in human survivorship curves, which has similarly been suggested to result from reductions in extrinsic mortality (through medical advancements and decreased physical hazards) [49], rather than from changes in intrinsic mortality patterns.

Besides lifespan, life history traits can exhibit significant diversity within a population, which is crucial for the resilience, productivity, and evolvability of populations [50–52]. However, empirical studies of life history at the population level often face challenges in measuring within-population diversity or detecting the presence of exceptional subpopulations, such as centenarians [53]. AEGIS provides direct access to individual-level intrinsic life history traits, revealing considerable standing genetic variation in the simulated populations (S3b and S3c Fig), even when variant phenotypes are rare. High genetic variation suggests that rapid adaptation may be possible, though intense bottlenecking could allow poorly adapted phenotypes to dominate.

## AEGIS simulates population dynamics

Previously, in Figs 3 and 4, we presented demography as static cross-sectional snapshots; however, populations in AEGIS are inherently dynamic, as are natural populations (Fig 5). These dynamics arise from the interplay between evolved intrinsic traits, i.e., mortality and fertility, and extrinsic factors, such as limited resources. Population dynamics critically influence the evolution of life history traits [9,54,55]. In AEGIS, population dynamics emerge spontaneously but can also be explicitly modified to investigate their causal relationship to life history evolution.

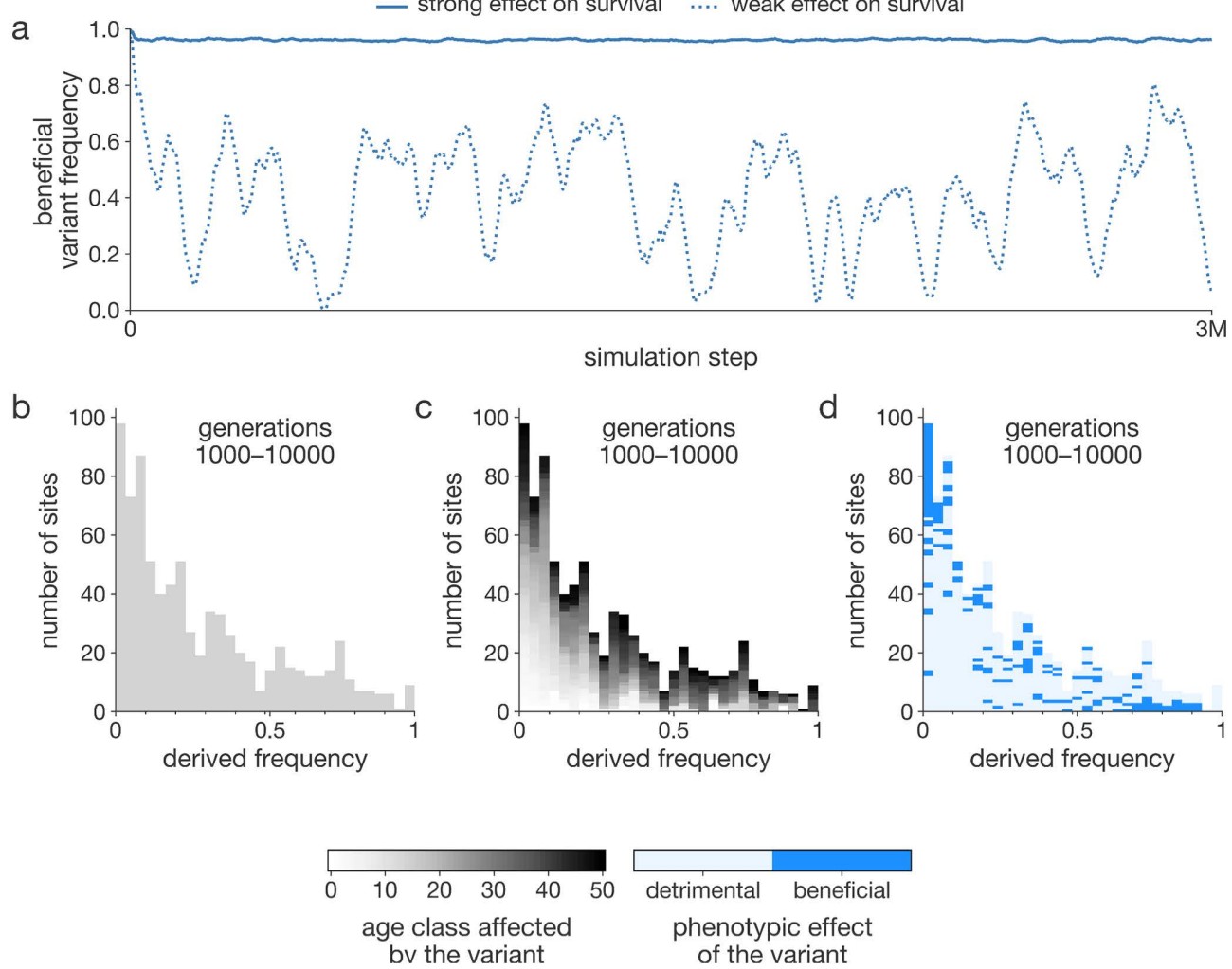

**Fig 6. Variant frequency analysis of an evolving population.** AEGIS enables the analysis of variant frequencies, which can be done either individually (a) or summarized through site frequency spectra **(b-d)**. a, Frequencies of two survival-enhancing variants with differing effect sizes are shown, with the stronger variant maintained at a higher frequency. b, Site frequency spectra (SFS) can be used to examine evolutionary forces, comparing generation 1000 (ancestral) with generation 10,000 (derived), showing a pattern consistent with neutral evolution. c, Color coding each variant by the age class it affects reveals that early-acting variants are overrepresented at low frequencies. d, Shading each variant by fitness effect shows early-acting (dark blue), low-frequency variants are primarily deleterious, while high-frequency variants are predominantly beneficial.

Periodic extrinsic pressures are recognized as influential forces in shaping the evolution of aging, although their effects can be challenging to model accurately [56]. We implemented a module mimicking abiotic factors of mortality which are periodic (Fig 5c). We can study continuous (S4a-c Fig) and intermittent pressures (Fig 4e and 4f) resulting in cyclical and periodic fluctuations of the size of the evolving population.

How populations respond to resource limitation strongly shapes demography and, consequently, the evolution of life history [57]. Population response to resource availability is marked by significant physiological diversity [58–68], which manifests in varying mortality patterns at the demographic level—patterns that we aim to capture within the AEGIS framework. By default, the mortality in response to starvation is delayed and increases gradually (Fig 5b); however, the delay

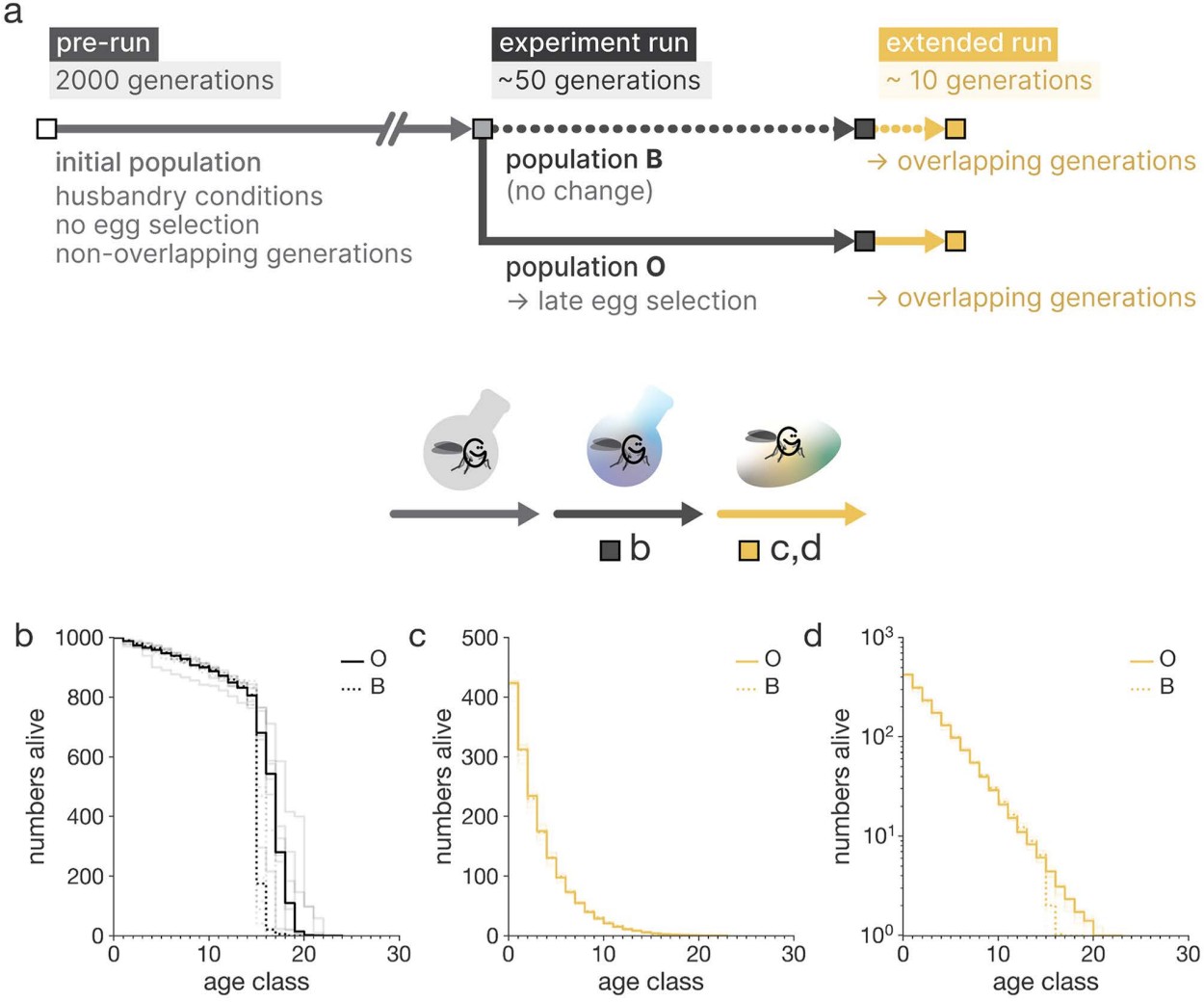

**Fig 7. A practical experimental example in AEGIS. a, Experimental design that mimics conditions in the Rose's experiment.** The panel lays out three phases of the design, with simulation length, evolutionary conditions which bifurcate for the two Rose's populations, B and **O**. The extended run is the addition to the experiment which simulates the evolved life histories in a wild-like environment. Carrying capacity for all simulations is 1000. We show life history from birth; maturity is obtained at age class 14. b, Survival curves of the baseline population B, and the population O with postponed senescence after 50 generations of evolution. Both median and maximum lifespan is greater in O. c, Survival curves of the two evolved populations in an environment with overlapping generations after 10 generations. No differences in population survival are discernible. d, Same as panel c, on a log-linear scale. Differences in population survival are discernible in late-life.

can be adjusted using input parameters. In an illustrative, extreme scenario with no delay, the population shrinks immediately down to carrying capacity when confronted with starvation (S4d Fig).

## AEGIS supports population genetics analyses

AEGIS offers access to simulated genotypic data across individuals and simulation time (Fig 6a). While the simulated data do not include molecular information or references to specific genes, they describe a set of variants with defined effects on life-history traits. This allows users to identify loci influencing the evolutionary dynamics of these traits.

Simulated genotypic data can be summarized and analyzed using population genetic methods, which are vital for extracting insights from large datasets. To explore the evolutionary forces—such as directional selection, balancing selection, or neutral processes—driving life-history evolution in each simulation, one might employ the site frequency spectrum (SFS). The SFS summarizes allele frequencies within a population and is widely used to infer evolutionary processes, including selection, genetic drift, and demographic changes. In Fig 6b, we present the SFS for the ancestral population at generation 1000 and the derived population at generation 10,000. Visual inspection reveals that the SFS lacks the characteristic U-shaped curve, suggesting the absence of a selective sweep within this timeframe. Additionally, the absence of an excess of intermediate-frequency alleles suggests that balancing selection did not occur. These patterns indicate that evolution may have proceeded primarily under neutral drift or weak background selection.

Given that the phenotypic effects of each variant are explicitly defined in AEGIS, this information can be incorporated into site frequency spectra analyses. For instance, it is possible to examine the age at which a variant exerts its effect (Fig 6c) or whether the variant has beneficial or deleterious consequences (Fig 6d). This allows for more detailed investigations into the evolutionary dynamics of trait-associated variants, providing deeper insight into how these variants shape life-history evolution.

## AEGIS: A practical example

To illustrate the application of AEGIS for *in silico* experimental evolution and its potential capabilities, we replicated and extended the foundational experiment on evolution of aging conducted by Rose [41] (from here on referred to as Rose's experiment), which tested whether imposing delayed reproduction could drive the evolution of postponed senescence in laboratory-maintained Drosophila populations.

We configured the experimental settings – parameterization, initialization, and termination – to best replicate the conditions of Rose's study (Fig 7a). Our simulation involves the evolution of two populations with non-overlapping generations – following Rose's original notation, a base population (B) and a population (O) in which only the oldest individuals reproduce. Both populations share identical simulation parameters, with the only difference being that in the O population, individuals can reproduce beyond day 14 (up to 50 days) and the eggs laid later were preferentially carried over into the next generation. In Rose's study, population B was maintained for 50 generations and population O for 15 generations, whereas both of simulated populations were run for approximately 50 generations (equivalent to 1000 steps, with the number of generations contingent on cohort lifespan). We initialized the B and the O population using a pre-evolved population that evolved under B conditions but for a significantly longer time (over 2000 generations) to prevent initialization artifacts (see Methods: Experimental Design).

The simulations show that population O evolves both an increased median and maximum lifespan (Fig 7b), suggesting that evolutionary pressures favor the evolution of extended lifespan, consistent with the experimental findings of Rose.

Although our experiment indicates that selection for late-life reproduction contributes to the evolution of delayed senescence, it remains unclear whether other conditions of the experimental evolution are necessary as well. Importantly, several aspects of the experimental setup were not varied, meaning that the experiment does not test whether these aspects are essential for the evolution of delayed senescence.

Using AEGIS, we can examine the importance of different conditions for the evolution of delayed senescence. One critical difference between laboratory and wild-like scenarios is the high level of extrinsic mortality present in the wild. It has long been posited that high extrinsic mortality obscures the selection for late-life traits in natural populations. We can assess how our findings generalize to such environments by simulating a wildlife scenario through two key modifications: transitioning from non-overlapping to overlapping generations and introducing limited resources, which impose extrinsic mortality on the population. In the wild-like context, we do not study the evolution of senescence *per se*; instead,

we observe how evolved populations perform under these wild-like conditions. Our results show that the evolved populations in wild-like conditions exhibit minimal differences in survivorship between the O and B populations, with most of the survivorship curves being nearly identical (Fig 7c) and only slight variation in late-life survival (Fig 7d). This suggests that the fitness advantage of longer lifespan observed in the laboratory is not strongly expressed in wild-like environments and indicates that such adaptations are unlikely to evolve under wild conditions. This highlights how AEGIS can effectively generalize findings which may otherwise be difficult to achieve.

## Discussion

Research on life history evolution has produced a variety of analytical and computational models to study the evolution of survival and reproduction. While these approaches have yielded substantial insight, many are optimized for analytical tractability or specific modeling goals, which can make it challenging to explore stochasticity, individual variation, and emergent dynamics within a single framework. AEGIS builds on this body of work by providing a powerful computational platform for simulating the evolution of life history traits, such as age-dependent mortality and fertility, across diverse environmental conditions. This model allows researchers to study the spontaneous emergence and evolution of aging traits, integrating various forms of extrinsic mortality and enabling a more nuanced exploration of the mechanisms driving aging and longevity. By facilitating a deeper understanding of these dynamics, AEGIS significantly enhances the toolkit available for life history and aging research.

AEGIS distinguishes itself from traditional analytical methods by its ability to simulate complex, emergent phenomena while relaxing some of the simplifying assumptions typically made by these methods. While analytical models often focus on population averages and deterministic outcomes, AEGIS captures the dynamic interactions between individuals and their environment, allowing for the evolution of life history traits to emerge naturally from these interactions. This approach enables the modeling of stochastic events, individual variation, and rare phenotypes, providing insights into evolutionary dynamics that analytical methods might overlook or find challenging to represent. Additionally, accessibility in AEGIS to researchers without advanced mathematical training broadens its utility, making it a valuable tool for exploring evolutionary questions that are difficult to address using traditional mathematical models. At the same time, agent-based models like AEGIS require careful interpretation, as they do not offer the same formal analysis as traditional models. While analytical approaches rely on mathematical formalism, agent-based models must use empirical validation through controls, replicated simulations, and statistical analysis. Thus, while AEGIS provides valuable insights, its results must be carefully contextualized through rigorous model design and validation.

Compared to other agent-based models, AEGIS is specifically tailored for studying life history evolution, integrating genotype-phenotype relationships to reveal how genetic variation shapes life history traits. Unlike more generalized ABMs, AEGIS allows these traits to emerge from the interplay of genetic, ecological, and demographic factors, offering a level of detail and focus that is essential for life history research. Its built-in analytical tools and user-friendly interface, including web-based accessibility, set AEGIS apart as a versatile and powerful platform.

AEGIS is designed to focus on the evolutionary forces shaping life history traits such as mortality and fertility, deliberately choosing to exclude certain features typical of some agent-based models, such as individual learning, decision-making, and sensory behaviors. This design choice prioritizes modeling fitness outcomes rather than delving into the molecular and behavioral processes underlying adaptation.

Building on its capabilities, AEGIS enables novel exploration of key questions in life history and aging evolution. It facilitates investigation into conditions for lifespan evolution, the role of genetic drift, and the performance of evolved life histories in new, challenging environments. Additionally, AEGIS supports studying the effects of environmental change on aging, the impact of different genetic architectures, and the evolutionary drivers of aging acceleration and deceleration. By offering simulated phenotypic and genotypic data, AEGIS provides a new perspective on these fundamental issues, advancing our understanding of evolution of aging and life history.

While AEGIS does have areas for future enhancement, these limitations offer exciting opportunities for development. While the current codebase could accommodate spatiality, its absence presently restricts AEGIS's capacity to simulate group selection—an element that may prove crucial for investigating theories of longevity and programmed aging. The inclusion of currently missing parental care mechanisms also presents an opportunity to deepen our understanding of longevity and human life history including hypotheses such as the grandmother hypothesis. Additionally, revising the assumption of uniform resource consumption could improve insights into how starvation and resource scarcity impact demographic patterns, which are central to theories like the disposable soma theory. Though the computational demands are generally manageable, further refinements could mitigate limitations associated with mutational meltdown in undersized populations and enhance longterm evolutionary analyses. Lastly, expanding AEGIS customization, particularly the order of computational processes, would further help test model assumptions and improve result robustness.

## Supporting information

**S1 Fig. Observed life history traits of evolved populations, over generations.** Average population-level observed survivorship, mortality and fertility curves at generation 50 (a-c), 250 (d-f) and 20000 (g-i). At generation 50, no age-dependent pattern in mortality nor fertility is discernible. At generation 250, observed mortality increases with age, while observed fertility decreases with age. At generation 20000, same trends are visible, but they are more intense. Furthermore, they are indistinguishable from phenotypes at generation 200000 (Fig 4a-4c).
(TIF)

**S2 Fig. Intrinsic life history traits of evolved populations, over generations.** Average population-level intrinsic survivorship, mortality and fertility curves at generation 50 (a-c), 250 (d-f) and 20000 (g-i). At generation 50, no age-dependent pattern in fertility is discernible and mortality is extremely low. At generation 250, observed mortality increases with age, while observed fertility weakly decreases with age. At generation 20000, age-specific trends emerge which are also much more intense. Furthermore, they are indistinguishable from phenotypes at generation 200000 (Fig 4d-f).
(TIF)

**S3 Fig. Intrinsic life history traits of individuals in evolved population.** The life history curves shown are based on 100 simulations which ran for three million steps. Thick curves are population averages. Thin lines are individual-specific traits. Plotted traits are survivorship (a), mortality (b) and fertility (c). Survivorship is commonly understood as an observed trait of a cohort rather than an intrinsic trait of an individual; however, we include it since it is visually informative. The technical interpretation of individual intrinsic survivorship is the survivorship of a hypothetical cohort of genetically identical individuals that die only due to intrinsic causes.
(TIF)

**S4 Fig. Population dynamics under various regimes of starvation and weather.** Six simulations with six different extrinsic mortality regimes. Carrying capacity in all simulations is 1000. The shaded areas depict the population size during a single step; the bold lines are averages. The first three panels (a-c) show populations that exhibit different sensitivities to starvation, i.e., different increases in mortality under lack of resources. Population a is most starvation-resilient, while the population c is least starvation-resilient. Populations d responds to overshooting precisely, thus do not dip below the carrying capacity. Populations e and f experience bursts of periodic abiotic mortality (every 25 steps) of random magnitude. Population e responds to overcrowding precisely, while population f suffers under overcorrection.
(TIF)

**S5 Fig. Evolving mutation rate.** The evolved mutation rates shown are based on 100 simulations which we ran for three million steps. Environment is stable; i.e., fitness of each phenotype does not depend on simulation time. a,

Population- and time-averaged mutation rate for populations evolving under different sizes. b, Population-averaged mutation rate over the simulation time. The darker line represents the average mutation rate of the largest population (K = 2892), the lighter line of the smallest population that did not go extinct (K = 516). Populations are reproducing asexually.
(TIF)

**S6 Fig. Evolution of intrinsic life history traits during the simulation.** The life history curves shown are based on 100 simulations we ran for three million steps. Red represents phenotypes recorded early in the simulation, violet late. For simulations depicted in panels a and b, mortality was initialized as high, while in panels d and e, it was initialized as low. Evolved states (in violet) do not differ significantly. For simulations depicted in panel c, fertility was initialized as low, while in panel f it was initialized as high. Evolved fertilities (in violet) do not differ significantly.
(TIF)

**S1 File. Supplementary information.**
(DOCX)

## Acknowledgments

We thank all members of the Valenzano lab over the years for the continuous feedback and key discussions that helped improve AEGIS. We are particularly thankful to Erik Boelen Theile for his key contribution to developing SFS in AEGIS during his Masters work in our group. This project greatly benefited from the many discussions had with colleagues from the DFG Collaborative Research Center "Predictability in Evolution" (SFB1310).

## Author contributions

**Conceptualization:** Arian Šajina, Dario Riccardo Valenzano.

**Data curation:** Martin Bagic.

**Formal analysis:** Martin Bagic, William John Bradshaw.

**Funding acquisition:** Dario Riccardo Valenzano.

**Investigation:** Martin Bagic, Arian Šajina, William John Bradshaw, Dario Riccardo Valenzano.

**Methodology:** Martin Bagic, William John Bradshaw, Dario Riccardo Valenzano.

**Project administration:** Dario Riccardo Valenzano.

**Resources:** Dario Riccardo Valenzano.

**Software:** Martin Bagic, Arian Šajina, William John Bradshaw.

**Supervision:** Dario Riccardo Valenzano.

**Validation:** Martin Bagic.

**Visualization:** Martin Bagic.

**Writing – original draft:** Dario Riccardo Valenzano, Martin Bagic.

**Writing – review & editing:** Dario Riccardo Valenzano, Martin Bagic.

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
