## [Decision Letter · Decision Letter 0]

28 Nov 2025

PCOMPBIOL-D-25-01268

AEGIS: individual-based modeling of life history evolution

PLOS Computational Biology

Dear Dr. Valenzano,

Thank you for submitting your manuscript to PLOS Computational Biology. After careful consideration, we feel that it has merit but does not fully meet PLOS Computational Biology's publication criteria as it currently stands. Therefore, we invite you to submit a revised version of the manuscript that addresses the points raised during the review process.

We look forward to receiving your revised manuscript.

Kind regards,

Ricardo Martinez-Garcia

Academic Editor

PLOS Computational Biology

Natalia Komarova

Section Editor

PLOS Computational Biology

**Additional Editor Comments:**

I agree with both reviewers that your work provides a very useful tool to perform mechanistic simulations of aging dynamics. However, I agree with Reviewer 2 that the most impactful part of your work is the method itself. Therefore, I recommend that you submit a revised version of this work, addressing all the comments raised by the reviewers, as a Methods paper.

**Journal Requirements:**

At this stage, the following Authors/Authors require contributions: Martin Bagic, Arian Šajina, William John Bradshaw, and Dario Riccardo Valenzano. Please ensure that the full contributions of each author are acknowledged in the "Add/Edit/Remove Authors" section of our submission form.

**Reviewers' comments:**

Reviewer's Responses to Questions

**Comments to the Authors:**

Reviewer #1: Bagic et al., propose a software suite for simulating the evolution of life history traits.

Their proposed solution surely makes it relatively easy to run simulations for these “ageing evolution” models. It is very easy to use through the web server as well as either through the pip install that went smoothly. In addition, the visual representations are easily understandable.

As mentioned by the authors lines 73-75, IBM are bottom-up models that allow emergent behaviors instead of needing to implement specific rules. However, numerous assumptions made for the implementation of the model (i.e. the sequence of events affecting an individual) seem to create unevaluated implicit constraints.

The implementation of the calculations methods themselves seem to be based on “An In Silico Model to Simulate the Evolution of Biological Aging”, double posted on arxiv and biorxiv for which I could not find a peer-reviewed version. The original paper does not seem to be comparing the proposed model with existing ones making it hard to understand what the conceptual or technical advances come along with this new implementation. For example, how does the current implementation of the code execution time compare with existing methods.

Some methodological choices are not clearly explained, i.e. how does continuous age of individuals vs discrete time steps interact here; how are mutating genes selected. Are they all mutated at every step, sampled independently?

p.11 parallelization role is not clear here since, for example, generating the t+1 population from t without modifying t would give the same effect. In addition, p.12 the description of interactions “modifying the pool of available mates for other individuals” is hard to comprehend if the model is truly parallelized.

I really appreciated the clever implementation of the isolation of an evolved population after burn-in for testing the impact of specific parameters, although this limits the study of evolutionary scenarios on stationary populations only. However, this limits the interpretation of parameters’ role to the context of stationary populations only. I know that this is a general assumption of population genetics but numeric simulations should be able to give us insights that are not or less accessible due to assumptions made for mathematical modeling. The authors could maybe show the parameters compared with and without burn-in on models for which they know the expected outcome.

Implementing randomization of relevant processes during each step might give significant (see comments from the authors page 10) improvement of outcomes relevance. i.e. averaging results between sims obtained with same parameters but infection / predation inverted. More generally, as stated by the authors lines 340-343, “not all simulations in AEGIS will necessarily exhibit a flat pre-reproductive mortality” implies that drawing conclusions from simulations should require the implementation of averaged outcomes.

Definition of intrinsic mortality define as “truly independent of external factors”

Starvation mortality being independent of genetics is a strong choice.

Why only one progeny / reproduction cycle?

I regret that there is no further discussion of the final result presented as an example in Fig S.6

Discussion starts by mentioning the lack of comprehensive tools, which is not true as indicated by the list of existing solutions mentioned by the authors in introduction. Even though they do not have a GUI, they have clear publications backing them.

Mentions to late-life plateau argument page 16 and the relative convergence with the Rose experiment are hard to judge since the late-life mortality plateau is debated – and using the model to propose explanations would have been interesting – and the Rose selection shows a relatively low impact on the lifespan. Comparing the outcome with other models while showing the value added by AEGIS on interpretability would have been more relevant.

p. 23 lines 479-481 is an overstatement as 1) numeric simulations only provide empirical testing of hypothesis that will never replace formal analysis of models and 2) it gives an illusion of simplicity where, the large number of parameters make the model difficult to interpret even though the authors implemented a method for isolating a stationary population to play with parameters on.

Reviewer #2: This submission proposes a new computational tool (AEGIS) to run agent-based simulations in Python with some of the most typical parameters that are of general interest to research in evolutionary biodemography. I believe the proposed tool is potentially interesting. However, I am unsure as to the standards by which this contribution should be judged. AEGIS is a computational method/tool and not a result per se. Whether the proposed tool will gain traction and deliver new results is up to future research. The manuscript contains illustrations of how AEGIS simulations are coherent with a number of relevant observations about life histories. This is a very good sanity check, however, it does not per se constitute a novel research result. I guess I would leave it to the editor to assess how this submission actually conforms to the article type “Research article” for PLoS CB.

That said, there are a few features of AEGIS I would like to comment upon as they are susceptible of improvement:

1. The authors choose for parallel execution so that individuals execute their actions independently of another individuals as opposed to have one individual executing after the other. I guess there are context where it makes sense to have both possibilities open, as they are both realistic in different scenarios.

2. Mortality/fertility appear to be specified as either an age-independent constant (this gives you a hold over potentially infinite ages) or a vector of age-specific values (this is limited to ages as numerous as the vector length). It would be ideal to have the possibility to have mortality/fertility functions that allow you to specify a different value for any age (possibly infinite)

3. The authors note that (lines 330-40) that in their simulations aging starts with sexual reproduction. Please note that this replicates a theoretical result by Hamilton 1966.

4. A major factor that can influence the analysis of life histories as emerging from individual trajectories is demographic heterogeneity, i.e., different subsets of individuals follow different life histories. I find unclear if and how AEGIS deals with demographic heterogeneity when reporting the simulation results.

5. I would like to have a more detailed description of how environmental variability is introduced. Can one for example implement a Markov chain that specifies the environmental state? It seems the key implementable process is some logistic growth (carrying capacity, K).

Typos:

Line 75: “assumed and built into the.”

**Have the authors made all data and (if applicable) computational code underlying the findings in their manuscript fully available?**

Reviewer #1: Yes

Reviewer #2: Yes

PLOS authors have the option to publish the peer review history of their article (what does this mean? ). If published, this will include your full peer review and any attached files.

**Do you want your identity to be public for this peer review?** For information about this choice, including consent withdrawal, please see our Privacy Policy .

Reviewer #1: No

Reviewer #2: No

**Figure resubmission:**
---

## [Decision Letter · Decision Letter 1]

9 Mar 2026

Dear Dr Valenzano,

We are pleased to inform you that your manuscript 'AEGIS: individual-based modeling of life history evolution' has been provisionally accepted for publication in PLOS Computational Biology.

Best regards,

Ricardo Martinez-Garcia

Academic Editor

PLOS Computational Biology

Natalia Komarova

Section Editor

PLOS Computational Biology

Reviewer's Responses to Questions

**Comments to the Authors:**

Reviewer #1: I thank the authors for their responses and clarification.

Reviewer #2: The authors have replied to my comments that my suggestions for improvement may be part of some future version of the software. I find this reasonable.

**Have the authors made all data and (if applicable) computational code underlying the findings in their manuscript fully available?**

Reviewer #1: Yes

Reviewer #2: Yes

PLOS authors have the option to publish the peer review history of their article (what does this mean? ). If published, this will include your full peer review and any attached files.

**Do you want your identity to be public for this peer review?** For information about this choice, including consent withdrawal, please see our Privacy Policy .

Reviewer #1: **Yes:** Michael Rera

Reviewer #2: No

---

## [Editor Report · Acceptance letter]

PCOMPBIOL-D-25-01268R1

AEGIS: individual-based modeling of life history evolution

Dear Dr Valenzano,

I am pleased to inform you that your manuscript has been formally accepted for publication in PLOS Computational Biology. Your manuscript is now with our production department and you will be notified of the publication date in due course.

With kind regards,

Anita Estes
